# Distinct Whole Transcriptomic Profiles of the Bursa of Fabricius in Muscovy Ducklings Infected by Novel Duck Reovirus with Different Virulence

**DOI:** 10.3390/v15010111

**Published:** 2022-12-30

**Authors:** Tao Yun, Jionggang Hua, Zheng Ni, Weicheng Ye, Liu Chen, Yinchu Zhu, Cun Zhang

**Affiliations:** State Key Laboratory for Managing Biotic and Chemical Threats to the Quality and Safety of Agro-Products, Institute of Animal Husbandry and Veterinary Sciences, Zhejiang Academy of Agricultural Sciences, Hangzhou 310021, China

**Keywords:** novel duck reovirus, bursa of Fabricius, whole transcriptomic analysis, miRNA

## Abstract

Novel duck reovirus (NDRV) is a newly identified reovirus that brings about more severe damage on multiple organs and mortality in various species of waterfowl. We previously characterized the transcriptomic profiles responding to NDRV in the bursa of Fabricius of Muscovy ducklings, which is a major immunological organ against virus infection. However, the molecular mechanisms of variant cell responses in the bursa of Fabricius to NDRV with different virulence is unclear. Here, we conducted a whole transcriptomic analysis to study the effects of two strains, HN10 (virulent NDRV) and JDm10 (artificially attenuated NDRV), on the bursa of Fabricius of Muscovy ducklings. We harvested a large number of differentially expressed genes (DEGs) of the bursa of Fabricius specially induced by HN10 and JDm10, and we found that HN10 induced DEGs enriched in differentiation and development in multiple organs beyond JDm10. Moreover, the ceRNA regulatory network also indicated the different connections among mRNA, lncRNA and miRNA. Interestingly, we further noticed that a population of differential expressed miRNA could particularly target to transcripts of HN10 and JDm10. We took miR-24 as an example and observed that miR-24 could reduce the transcription of GLI family zinc finger 3 (Gli3) and membrane-associated guanylate kinase, WW and PDZ domain containing 1 (Magi1) via recognition 3′ UTR of these two genes by a dual luciferase reporter gene assay in vitro. However, this effect could be compromised by HN10 infection or the ectopic over-expression of the putative miR-24 targeting regions in L1 and L3 fragments of HN10. Taken together, we examined and proposed a novel regulatory competitive mechanism between transcripts of NDRV and Muscovy ducklings for miRNA. These findings may advance the understanding of the molecular pathogenesis of NDRV in Muscovy ducklings, and help provide the potential targets for vaccine and drug development against NDRV.

## 1. Introduction

Duck reovirus (DRV) is a common waterfowl virus with various morbidities, belonging to the genus Orthoreovirus in the family Reoviridae [1,2]. DRV infection in waterfowl has been found in many parts of the world, mainly in Europe and Asia [2,3]. Based on biological and genetic properties, DRV can be classified into two genotypes: genotype I (Classical DRV, CDRV) and II (Novel DRV, NDRV) [2,4]. So far, CDRV has only been identified in Muscovy duck (*Cairina moschata*) and goose (*Anser anser domestica*), and the disease is typically characterized by liver and spleen swelling with small white necrosis foci on the surface [4,5]. It also causes lameness, stunting and immune suppression in young waterfowl [4,6,7]. In the early 2000s, NDRV first emerged in China [4]. Subsequently, the disease broke out in major waterfowl-producing areas of China. NDRV has become the prevalent genotype in China. Compared with CDRV, NDRV can cause more severe symptoms, higher mortality and a wider host range [4,8,9,10,11]. The hosts infected with NDRV, particularly young birds (5–10 days old), exhibit the serious hemorrhaging and irregular massive necrosis in the liver and spleen, with a mortality of even 50% [12]. Furthermore, NDRV can infect almost all species of waterfowl, such as the Pekin duck (*Anas platyrhynchos domestica*) and Mallard duck (*Anas platyrhynchos*), besides Muscovy duck and goose. The emergence and epidemic of acute hemorrhagic necrotic hepatitis has brought about a serious economic loss to the waterfowl industry in China. However, there is no effective vaccine against NDRV infection.

Although CDRV and NDRV both belong to Avian orthoreovirus species group II, the sequence divergences of genes encoding the outer capsid protein, including μB, σB and σC, indicate the distinct genome structure, pathogenicity and antigenicity [4]. Moreover, the sequence variability of σC is also observed in NDRV strains with different virulence, suggesting a potential role in pathogenicity [13,14]. NDRV has been identified as an immunosuppressive pathogen, which can cause various degrees of damage to immune organs after infection, especially to the spleen and bursa of Fabricius, and thereby affect the immune function of the host [11,15,16,17]. The virulent NDRV can cause severe hemorrhages in the bursa of Fabricius, and lead to necrosis of most follicular lymphocytes, especially in the medullary region of follicles [11]. Lymphocytes not only show the membrane lysis, organelle extravasation and focal lytic necrosis, but also show chromosome aggregation and pyknosis. In turn, naturally or artificially attenuated NDRVs usually do not cause any clinical signs but only cause slight gross and pathological lesions, or even no lesions in ducklings. Nonetheless, the molecular pathogenic and immunomodulatory mechanisms between NDRVs with different virulence and hosts are not clear.

HN10 and JDm10 were two NDRV strains with stable virulence isolated from our lab. The virulent strain HN10 was first identified from a hemorrhagic and necrotic Muscovy duck liver and passaged in chicken embryo fibroblast (DF-1) cells for 20 generations [1,4,18]. JDm10 was obtained from the liver of a Muscovy duckling with a few white necrotic spots of smaller size and further attenuated by passage in DF-1 cells for 150 generations. JDm10, as a laboratory-made attenuated NDRV, had been determined to have less damage to the bursa of Fabricius than HN10 in our previous studies [15,19]. However, the differential changes of mRNA, miRNA, lncRNA and circRNA in bursa of Fabricius infected with differently virulent NDRVs have not been studied, and no detailed analysis of the correlation between ncRNA and mRNA levels has been performed.

In present study, we continued the whole transcriptome sequencing to investigate the different responses of the bursa of Fabricius of ducklings to HN10 (virulent NDRV) and JDm10 (artificially attenuated NDRV) infection. Our data are served as a resource for improving the understanding of the effects of NDRV on immunosuppression in waterfowl and is of significant benefit to vaccine design and the development of antiviral drugs.

## 2. Materials and Methods

### 2.1. Ethical Statement

All experiments were performed in the Institute of Animal Husbandry and Veterinary Sciences (IAHV), Zhejiang Academy of Agricultural Sciences (ZAAS). All animal samples were treated in accordance with the Regulations of the Administration of Affairs Concerning Experimental Animals approved by the State Council of China. The bird protocol used in this study was approved by the Research Ethics Committee of ZAAS (permit number: 2021ZAASLA09). One-day-old healthy Muscovy ducklings were used for the virus infection experiment.

### 2.2. Cells and Virus

DF-1 and duck embryo fibroblast (DEF) cells (CCL-141) were maintained in Dulbecco’s modified Eagle’s medium (DMEM) (Gibco, Shanghai, China) supplemented with 10% fetal bovine serum (FBS) (Gibco, Thermo Scientific, Grand Island, NY, USA), 100 IU/mL of penicillin and 100 μg/mL of streptomycin at 37 °C in a 5% CO_2_ incubator. The NDRV strains HN10 (virulent NDRV) and JDm10 (artificially attenuated NDRV) used in this study have been described previously [15].

### 2.3. Animal Study

The experimental infection was conducted in 1-day-old healthy Muscovy ducklings, and the serum samples from each duckling were detected by RT-PCR and ELISA to ensure that they were not infected with NDRV [15]. Twenty-seven one-day-old healthy Muscovy ducklings were collected and randomly separated into three groups. Nine ducklings were administrated with 500 μL JDm10 strain at a titer of 10^8.2^ median tissue culture infective dose (TCID50) per mL via intramuscular injection; nine ducklings were administrated with 500 μL HN10 strain at a titer of 10^6.4^ median TCID50 per mL; the remaining nine ducklings were treated with sterile DMEM as a blank control. For each group, nine ducklings were randomly and equally divided into three independent subgroups as three biological replicates.

### 2.4. Histopathological and Terminal Deoxynucleotidyl Transferase-Mediated dUTP-Biotin Nick End Labeling (TUNEL) Analysis

The pathological changes of the bursa of Fabricius were observed with H&E staining. The bursa of Fabricius specimens were collected at 72 h post infection (hpi). Histological sections were routinely prepared from each group of bursa samples. The bursa samples were fixed in 10% neutral formalin for 48–72 h, embedded in paraffin, sectioned at a thickness of 5 µm, stained with hematoxylin and eosin (H&E) and evaluated using a light microscope (Eclipse Ci-L, Nikon, Tokyo, Japan). For analysis of NDRV-induced apoptosis, A TUNEL assay was conducted using an In Situ Cell Detection kit (Roche, Beijing, China) according to the manufacturer’s instructions. Nine sections were randomly selected from each group, and the number of positive brown cells (apoptotic cells) was counted for each section in three visual fields. Positive cells and total cells were counted by Image-Pro Plus 6.0 software. The formula used to evaluate apoptosis was apoptosis index = positive brown cells × 100%/total number of cells.

### 2.5. Whole Transcriptome Sequencing

The total RNA of the bursa of Fabricius tissues was extracted using RNAiso Plus (Takara Bio Inc., Dalian, China). The RNA amount and purity of each sample were quantified using NanoDrop ND-1000 (NanoDrop, Wilmington, DE, USA). The RNA integrity was assessed by Bioanalyzer 2100 (Agilent, Santa Clara, CA, USA) with RIN number >7.0, and confirmed by electrophoresis with denaturing agarose gel. The RNA was divided into two parts for at least 2 μg in each. An Epicentre Ribo-Zero Gold Kit (Illumina, San Diego, CA, USA) was used to deplete rRNA, and an NEB Next Ultra Directional RNA Library Prep Kit for Illumina (New England Biolabs, Ipswich, MA, USA) was used to establish the library of one part for long-chain RNA sequencing. The libraries were processed by the Illumina NovaseqTM 6000 platform (LC-Bio Technology Co., Ltd., Hangzhou, China) with paired-end 2 × 150 bp as the sequencing mode. A TruSeq Small RNA Sample Prep Kit (Illumina) was used to establish the library of the other part for small RNA sequencing. The libraries were processed by the Illumina Hiseq2000/2500 platform with single-end 1 × 50 bp as the sequencing mode.

For data analysis of long-chain RNA, raw data were processed for unqualified sequences removal by cutadapter. Clean data were mapped to reference genome (iascaas_pekingduck_pbh1.5, gcf_003850225.1) using Bowtie2 and Tophat2. The mapped reads were de novo-assembled circular RNAs by CIRCExplorer at first, then back splicing reads were identified in unmapped reads by Tophat-fusion and CIRCExplorer. All samples were generated unique circular RNAs. Circular RNA expressions from different samples or groups were calculated by SRPBM = (number of back-spliced junction reads)/(number of mapped reads) × 1,000,000,000. Only the comparisons with a *p* value less 0.05 were regarded as showing differential expression by R package edgeR.

For data analysis of small RNA, the unique sequences with a length of 18~25 nt were mapped to miRNA sequences in miRBase 22.0 (http://www.mirbase.org/). Mapping was also performed on pre-miRNA against genomic data. The unique sequences that aligned to the known miRNA sequences in miRBase 22.0 were identified as known miRNA. The secondary structure of pre-miRNAs was presented as a hairpin, including 5p- and 3p- derived miRNA. The unique sequences mapping to the other arm of the pre-miRNA sequences, which were not annotated in the miRBase 22.0, were considered to be 5p- or 3p-derived miRNA candidates. The remaining unmapped sequences were matched to the genomic sequences in search of candidate novel miRNAs. To identify the results of putative miRNAs, all the obtained miRNAs were used to predict the secondary structures using RNAfold software (http://rna.tbi.univie.ac.at/cgi-bin/RNAWebSuite/RNAfold.cgi). The miRNA differential expression based on normalized deep-sequencing counts was analyzed by selectively using ANOVA. The significance threshold was set to be 0.05 in the test. To predict the genes targeted by differentially expressed miRNAs, two computational target prediction algorithms (TargetScan 5.0 and miRanda 3.3a) were used to identify miRNA binding sites.

Finally, the data predicted by both algorithms were combined and the overlaps were calculated. The GO terms (http://www.geneontology.org/) and KEGG Pathway (http://www.genome.jp/kegg/) of these differentially expressed miRNA targets were also annotated. All raw data were deposited to the Arrayexpress database, assigned with the accession number E-MTAB-12115.

### 2.6. Dual Luciferase Reporter Gene Assay

The Dual Luciferase Reporter Gene Assay Kit (Yeasen, Cat. No. 11402ES60) was used for this experiment. A total of 5 × 10^5^ duck embryonic fibroblasts (DEFs) were subcultured in a 24-well plate. miR-24 mimics (5′-GGCTCAGTTCAGCAGGAACA-3′) were synthesized by TSINGKE (Shanghai, China), and transfected into DEFs. A 3′ UTR sequence of Gli3 and Magi1, as well as 2 miR-24-targeted sequences “AGGAGTTCCTTCAATCCTGGGCCT” (L1, 680–709) and “GTATGATCCTGAAATGGCAGCTGAGTCC” (L3, 1218–1245) of HN10 transcripts were respectively cloned into the downstream of the pGL3 firefly luciferase report vector (Promega, Madison, WI, USA). Additionally, these two miR-24 targeted sequences were also in vitro synthesized by TSINGKE. Vectors of pGL3 and phRL-TK (Rinilla luciferase) were co-transfected into DEFs using Lipofectamine 3000 (Thermo Fisher Scientific, Waltham, MA, USA) according to the manufacturer’s protocol. After HN10 or JDm10 induction for 12 h, the culture medium was removed, washed by PBS 3 times, added to 100 μL lysate for 30 min at room temperature and then mixed with 50 μL luciferase assay reagent II buffer. The intensity of firefly and rinilla luciferase was measured by a Multiskan FC microplate reader. The ratio of firefly and rinilla luciferase was used to evaluate the transcription activity.

### 2.7. Statistical Analysis

The statistical analysis was performed by using the Graph-Pad Prism (Prism 5.0, GraphPad Soft-ware Inc., San Diego, CA, USA), and data are shown as the mean ± standard error of mean (SEM). The statistical significance of the differences between two groups were calculated by a two-tailed Student’s *t*-test, and one-way ANOVA with the Turkey’s post hoc pairwise test was used for comparisons among multiple groups. Statistical significance was considered at *p* < 0.05.

## 3. Results

### 3.1. Histopathological Changes in NDRV-Infected the Bursa of Fabricius

Histopathological changes of the bursa of Fabricius infected by NDRV with different virulence are shown in Figure 1. In the virulent NDRV (HN10)-infected group, the H&E staining analysis of the bursa of Fabricius sections indicated there were significant histopathological changes, including marked necrosis and sloughing of epithelial cells in the mucosal; severe necrosis and depletion of lymphocytes in lymphoid follicles, and follicle atrophy; large amounts of necrotic cellular debris within the interfollicular interstitium with infiltration of neutrophils and lymphocyte, and proliferation of fibrous tissues (Figure 1A). In the attenuated NDRV (JDm10)-infected group, less lymphocyte necrosis was observed in the lymphoid follicles, the lymphocytes in the medulla were slightly reduced and arranged loosely and the infiltration of inflammatory cells was no obvious (Figure 1B). There were no evident histopathological lesions in the control group (Figure 1C).

In order to analyze the effects of NDRV infection on apoptosis, the TUNEL assay was used to detect the apoptosis of lymphocytes in tissue sections of bursa of different virulence NDRV infection (Figure 1D–F). The TUNEL assay showed that the number of apoptotic cells in the bursa of Fabricius was significantly increased in the HN10-infected group compared to that in the control group (*p* < 0.0001) and JDm10 infected group (*p* < 0.0001), respectively (Figure 1G).

### 3.2. Transcriptome Profile in the Bursa of Fabricius of Ducklings by NDRVs with Different Virulence

Initially, ducklings were inoculated with HN10 or JDm10 in vivo, and three groups of the bursa of Fabricius named “HN10”, “JDm10” and “Negative Control (NC)” were collected for transcriptome sequencing, including long chain RNAs (mRNA, lncRNA, circRNA) and short chain RNAs (mainly miRNAs) (please refer to “Materials and methods” for experimental details). We obtained total 783.33 M reads. The high ratio of mapped reads (Appendix A) and correlation coefficient among duplicate samples in each group (Appendix A) both indicate a good quality of specimens and sequencing.

Differentially expressed genes (DEGs) were analyzed. We harvested 1715 up-regulated genes including 1464 mRNAs, 208 lncRNAs, 42 miRNAs and one circRNA, as well as 407 down-regulated genes including 257 mRNAs, 88 lncRNAs, 59 miRNAs and 3 circRNAs compared between HN10 to NC (|log_2_ FC| >1, *p* < 0.05) (Figure 2A, Appendix A). Similarly, we also obtained 974 up-regulated genes including 830 mRNAs, 114 lncRNAs, 28 miRNAs and two circRNAs, as well as 183 down-regulated genes including 89 mRNAs, 52 lncRNAs, 41 miRNAs and one circRNA compared between JDm10 to NC (Figure 2B). The intersection of these DEGs shown by the Venn diagram accounted for 36.9% DEGs in HN10, while 67.6% DEGs in JDm10 compared to NC, suggesting that the effect of HN10 on gene expression in the bursa of Fabricius was more extensive than that of JDm10. These DEGs included 621 up-regulated and 46 down-regulated encoding genes (Figure 2C), 52 up-regulated and 26 down-regulated lncRNAs (Figure 2D), 11 up-regulated and 25 down-regulated miRNAs (Figure 2E) as well as one down-regulated circRNA named ciRNA57. Interestingly, the expression change of four lncRNAs was completely opposite between HN10 and JDm10. *LOC110351921* and *LOC106016928* were highly expressed in HN10, whereas reduced in JDm10 compared to NC. In turn, *LOC106016555* and *LOC110353382* were down-regulated in HN10, but elevated in JDm10 compared to NC (Figure 2F). Taken together, we profiled the differentially expressed gene of the bursa of Fabricius of ducklings by HN10 and JDm10 NDRVs.

### 3.3. DEGs Associated Biological Functions Affected by Different NDRVs

Next, the functions associated with DEGs were investigated. GO and KEGG analyses were separately conducted for specific DEGs of HN10 (SH) and specific JDm10 (SJ) as well as mutual DEGs of HN10 and JDm10 (SHJ) compared to NC. The top 20 GO terms with high confidence displayed that both HN10 and JDm10 NDRVs could affect the transmission process of signals from the extracellular microenvironment through cell adhesion and surface component into the cells in the bursa of Fabricius. Abnormal expression of *TLR4*, *STAB1*, *STAB2*, *RORA*, *SLC7A2*, *P2RX7*, *PLA2G1B*, *G2M*, *CD36*, *TNFSF8*, *C3AR1*, *ADIPOQ*, *TAP2* and *MMP28* finally cause the aberrance on macrophage activation and defense response to bacterium, and the vast majority of critical pathways such as Apelin, MAPK, PPAR, Hedgehog, Notch and Wnt signaling pathways were all affected (Figure 3A). Moreover, HN10-specific DEGs were also involved in multiple organs’ development including brain, heart, eye and uterus, platelet, bone, breast and liver (Figure 3B), indicating that multiple organ dysfunction and failure might be responsible for the high mortality of HN10. In turn, JDm10 was likely to affect neuron development and differentiation, as well as collagen production beyond other organs in ducklings (Figure 3C). Collectively, we characterized the distinct biological functions by NDRVs with different virulence.

### 3.4. Network Action Map of DEGs Based on the Sequence Targeting Analysis

Next, we further studied the genetic regulatory network in the bursa of Fabricius of ducklings by NDRVs. In terms of cis-regulatory modules, we summarized the adjacent mRNA and lncRNA of DEGs on chromatin. Nineteen pairs of mRNA-lncRNA interaction were discovered in the concurrent DEGs of SHJ compared to NC, whereas twenty-one pairs were found in DEGs of SH (Figure 4A). Different transcripts of the same gene showed the completely opposite correlation with the nearby lncRNA, indicating that lncRNA might participate in the alternative transcription initiation or splicing. No pair of mRNA and lncRNA was found in DEGs of SJ. Furthermore, in terms of competing endogenous RNA (ceRNA) module, we found that total 37 miRNAs connected 550 mRNAs and 89 lncRNAs in concurrent DEGs of SHJ compared to NC. Here, we paid attention to the biological function of the bursa of Fabricius and picked up the candidate genes associated with top 20 GO terms (biological functions). Our observations indicated different ceRNA networks in these three groups of DEGs, which contained eighty-five mRNAs, twenty miRNAs and five lncRNAs in DEGs of SHJ (Figure 4B), eighty-five mRNAs, twenty-nine miRNAs and eight lncRNAs in DEGs of SH (Figure 4C), as well as eighteen mRNAs, fifteen miRNAs and eight lncRNAs in DEGs of SJ (Figure 4D). Taken together, these results suggested the putative regulatory ceRNA modules in the bursa of Fabricius of ducklings by NDRVs.

### 3.5. Different miRNAs Targeted HN10 or JDm10

Since the network displayed that a small number of miRNAs could link with a large number of mRNAs and support the complicated networks of ceRNA, we suspected that the RNA of NDRVs could target certain miRNAs of ducklings and further largely affect expression of multiple mRNAs. To this end, we use miRanda tool to match miRNAs with transcripts of HN10 and JDm10. The given stem loop structures with high confidence across species showed the differentially expressed miRNAs targeting HN10 and JDm10 transcripts (Appendix A). Interestingly, the potential regions of HN10 transcripts targeted by 21 miRNAs were variant from those of JDm10 (Figure 5A). In turn, the regions of JDm10 transcripts targeted by 10 miRNAs were distinct from those of HN10 (Figure 5B). Eight miRNAs were found that let-7e-5p, miR-24, miR-30c-1-3p and miR-30e had at least two targeting sites in transcripts of HN10, whereas miR-146b-5p, miR-196b, miR-30c-2-3p and PC-5p-16192 had at least two targeting sites in transcripts of JDm10. We took miR-24 as an example. By RNA-seq, we found miR-24 could recognize 3′ UTR of GLI family zinc finger 3 (Gli3) and membrane-associated guanylate kinase, WW and PDZ domain containing 1 (Magi1) in the bursa of Fabricius treated with HN10, and also recognize the partial L1 and L3 of HN10 transcripts. To test our hypothesis of the competitive relationship between transcripts of NDRV and Muscovy ducklings for miRNA, a series of dual luciferase reporter gene assay were conducted. Figure 5C showed that miR-24 could significantly weaken the transcription activities of Gli3 and Magi1. Likewise, miR-24 could also target to two fragments of HN10 (Figure 5D). Then DEF cells were additionally treated with HN10 and JDm10. The fluorescence signal driven by 3′ UTR of Gli3 was compromised by miR-24 mimics but was rescued by HN10 infection and the ectopic expression of L1 (680–709) and L3 (1218–1245) fragments of HN10 (Figure 5E), which was similar with Magi1 instead of Gli3 (Figure 5F). JDm10 infection failed to affect the fluorescence signals.

Moreover, the target regions were mainly located in M-class and S-class segments of the HN10 genome (Figure 5G). The expression of each transcript of NDRVs in RNA-seq data of the bursa of Fabricius of ducklings was also evaluated. It was worth noting that the differential levels of each transcript in HN10 and JDm10 (Figure 5H) were negatively correlated with the miRNA counts (*r* = −0.627, *p* = 0.002). Collectively, we summarized the different miRNA profiles of the bursa of Fabricius of Muscovy ducks responding to HN10 and JDm10.

## 4. Discussion

NDRVs have been widely acknowledged to result in more symptoms of severe multiple organ failure than CDRVs in waterfowl. There have been several studies on whole transcriptome analysis of multiple organs responding to reovirus [15,20,21,22,23,24]. The transcriptomic data of our study fill the gap in the landscape of immune cell development in the bursa of Fabricius of Muscovy ducks. As a central immune organ for B-lymphocytes development unique to birds [25,26], the bursa of Fabricius starts to hatch B-cells at the embryonic stages [27], but the microenvironment of it that is really beneficial to B-cell maturation is at the neonatal stage [28,29,30]. Previous studies have determined that the spleen and bursa of Fabricius are the primary target organs by reovirus [17], and the consequent immunosuppression usually leads to the deficient B-cells unable to produce an antigen response in young waterfowls [31]. Nevertheless, another view indicates that an appropriate nonfatal and attenuated reovirus efficiently stimulates the ducklings to develop immunity immediately, which can protect from the infection and proliferation of virus strains with more virulence [13,15]. Our data provide a possible explanation for the balance behind this contradiction. The RNA profiles of the bursa of Fabricius responding to different virulent NDRV strains HN10 and JDm10 (Figure 1C–E) suggest that a large number of genes are abnormally regulated by HN10 and JDm10, respectively. NDRV-caused cell apoptosis [32], cell fusion [33] and membrane permeabilization [34] mentioned in previous studies are also found in our results. Beyond the concurrent functions and signaling pathways affected by NDRVs, the up/down-regulation of genes by HN10 closely connects with the dysfunction of differentiation and development of brain, heart, eye and uterus, platelet, bone, breast and liver (Figure 2C,D). Multiple organs failure appears to be the main cause of the high mortality of HN10 compared to JDm10. Our previous proteomic study determines that only immune-related proteins are induced by HN10 infection [15]. We suppose that they are not contrary. Transcript levels can reflect the tendency of most genes’ expression under stress [35]. For example, although the genetic changes in the bursa of Fabricius after NDRV stimulation are not indeed directly related to multiple organ functions, we can speculate that these genetic changes actually occur in multiple organs even though we only focus on the transcriptome of the bursa of Fabricius. However, the protein expression changes are the ultimate reflection of the bursa of Fabricius stimulated by NDRVs. Additionally, mass spectrometry is much less sensitive than sequencing in detecting differential changes. Therefore, the transcriptomic sequencing data in this study provide more information beyond the immune-related function.

To attempt to figure out the mechanism behind the different virulent effect of HN10 and JDm10 on the bursa of Fabricius, we incorporate the RNA of NDRVs into the analysis system. Avian reoviruses contain 10 double-stranded RNA (dsRNA) genome segments including three classes of designated L (large), M (medium) and S (small). Each segment can be further divided into individual regions for protein coding [36]. The genome alignment of HN10 and JDm10 indicates a large proportion of variations distributed everywhere between them. These variations may cause the changed codon optimization, alternative transcriptional start site and splicing sites, which all contribute to gene expression. Physically, viral RNA can interact with eukaryotic nucleotides as long as their sequencing meets the appropriate base-pairing conditions [37]. As a kind of non-coding RNA, miRNA can also affect the abundance and translation efficiency of viral mRNA [38,39]. This regulatory mechanism is not unusual although, and the coordination between mRNA of NDRVs and cellular miRNA of Muscovy duck is never illustrated. Now our analysis shows the specific miRNAs driven by HN10 or JDm10 (Figure 4A,B). We also find that the FPKM values of each segment of NDRVs are negatively correlated with the counts of the corresponding target miRNA, implying that cellular miRNAs of the bursa of Fabricius are likely to affect the proliferation of NDRVs.

Nevertheless, our analysis so far has no clue that whether the host ducklings take a direct or indirect approach against the viral mRNA. As a direct manner, the host can spontaneously enhance the expression of miRNAs to respond and degrade vital transcripts, specifically via complementary nucleotides and RNA-induced silencing complex (RISC) recruitment [40]. By sequencing alignment, most of the target regions on HN10 or JDm10 are neither located at 5′ or 3′-non-translated regions, which contribute to RNA stability, viral replication and translation. On the other hand, as an indirect manner, when the interferon (IFN) signaling cascade is induced by viral infection, this effect also extends to miRNA expression [40]. In other words, the complicated intracellular regulation responding to viral infection causes up-regulation of some miRNAs, and further unexpectedly affects viral RNA. Our ceRNA mode (Figure 4B–D) prompts that these miRNAs such as let-7e-5p, miR-24, miR-30c-1-3p, miR-30e, miR-146b-5p, miR-196b, miR-30c-2-3p and PC-5p-16192 are supposed to exert an inhibitory effect on protein translation of core elements of NDRVs. This hypothesis needs to be further addressed and verified by cellular and molecular study, although Gli3 and Magi1, as the target of miR-24 indicated by ceRNA mode, are never reported in other studies. MiR-24 indeed participates in controlling the immuno-related transcriptional signature [41]. In general, the sequence of non-coding RNA is not conserved across species. MiR-24 taken as an example, however, is almost identical in ducklings, mouse, rat and human species. The novel role of miR-24 in bridging mRNA and extraneous nucleotides in our research may also be true of other species for the studying of infection.

## 5. Conclusions

Our results have characterized the whole transcriptomic landscape of the bursa of Fabricius of Muscovy ducks infected by different virulence of NDRVs, and our analysis has indicated a novel interplay between viral RNA and miRNAs of Muscovy ducks for potential molecular mechanisms of host response. Our discovery is helpful for people to have a deeper understanding of the pathogenesis of NDRV and is expected to find effective targets to develop vaccines and drugs.

## Figures and Tables

**Figure 1 viruses-15-00111-f001:**
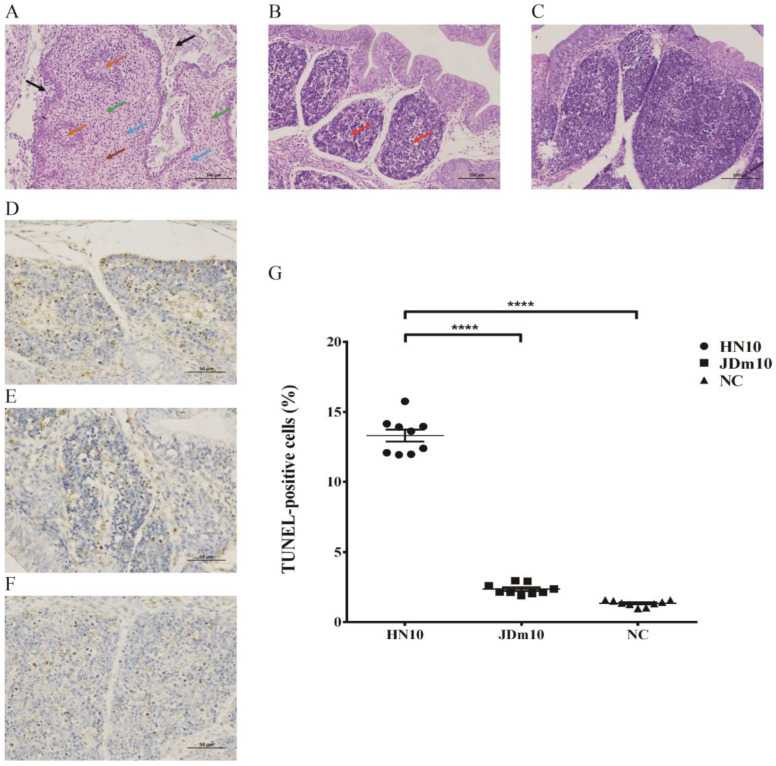
Microscopic lesions in the bursa of Fabricius of Muscovy ducklings inoculated with different virulent NDRV. (**A**–**C**) H&E staining shows the histopathological damage in the bursa of Fabricius infected by NDRV HN10 (**A**), JDm10 (**B**) and control (**C**) (Scale bar = 100 µm, Magnification 400×). Black arrows indicate the necrosis and sloughing of epithelial cells in the mucosal layer; orange arrows indicate the necrosis and depletion of lymphocytes in lymphoid follicles, and follicles atrophy; brown arrows indicate the large amounts of necrotic cellular debris; blue arrows indicate the interfollicular interstitium with infiltration of neutrophils and lymphocyte; green arrows indicate the proliferation of fibrous tissues. (**D**–**F**) TUNEL assay shows the measurement of apoptotic cells of the bursa of Fabricius infected by NDRV HN10 (**D**), JDm10 (**E**) and control (**F**) (Scale bar = 50 µm, Magnification 400×). (**G**) Percentage of apoptotic cells of the bursa of Fabricius infected with HN10 and JDm10. Data indicate mean ± SEM of 9 (control) and 9 ducklings (HN10, JDm10). One-way ANOVA with Bonferroni multiple comparison test, **** *p* ≤ 0.0001).

**Figure 2 viruses-15-00111-f002:**
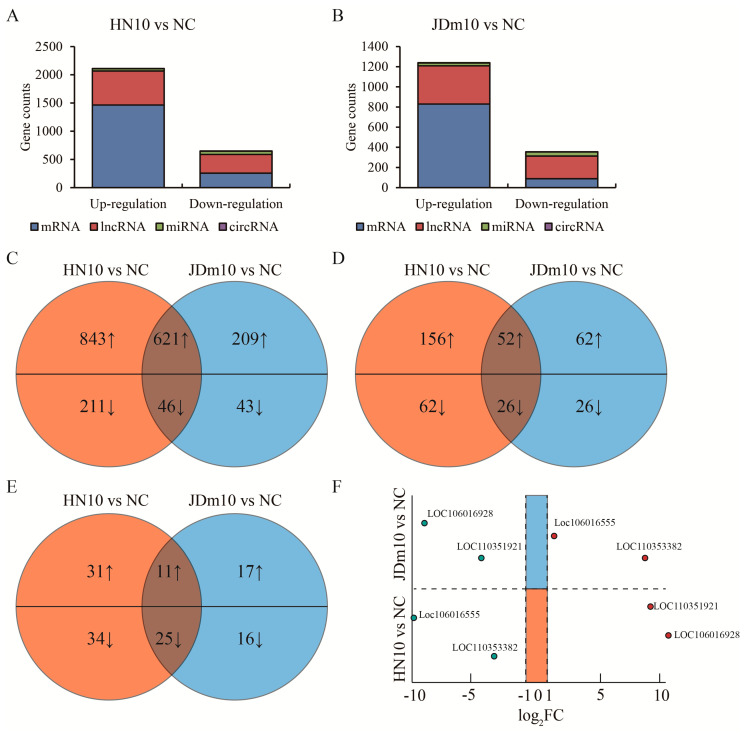
Analysis of mutual DEGs in the bursa of Fabricius induced by HN10 and JDm10. (**A**) Total DEG (|log_2_ FC| >1, *p* < 0.05) counts in the bursa of Fabricius treated with HN10 compared to negative control. (**B**) Total DEG counts in the bursa of Fabricius treated with JDm10 compared to negative control. (**C**–**E**) Venn diagrams showed the mutual up- or down-regulated mRNA (**C**), lncRNA (**D**) and miRNA (**E**) compared between HN10- and JDm10-induced DEGs. (**F**) Four lncRNAs were listed due to the opposite changes by HN10 and JDm10 treatment compared to negative control.

**Figure 3 viruses-15-00111-f003:**
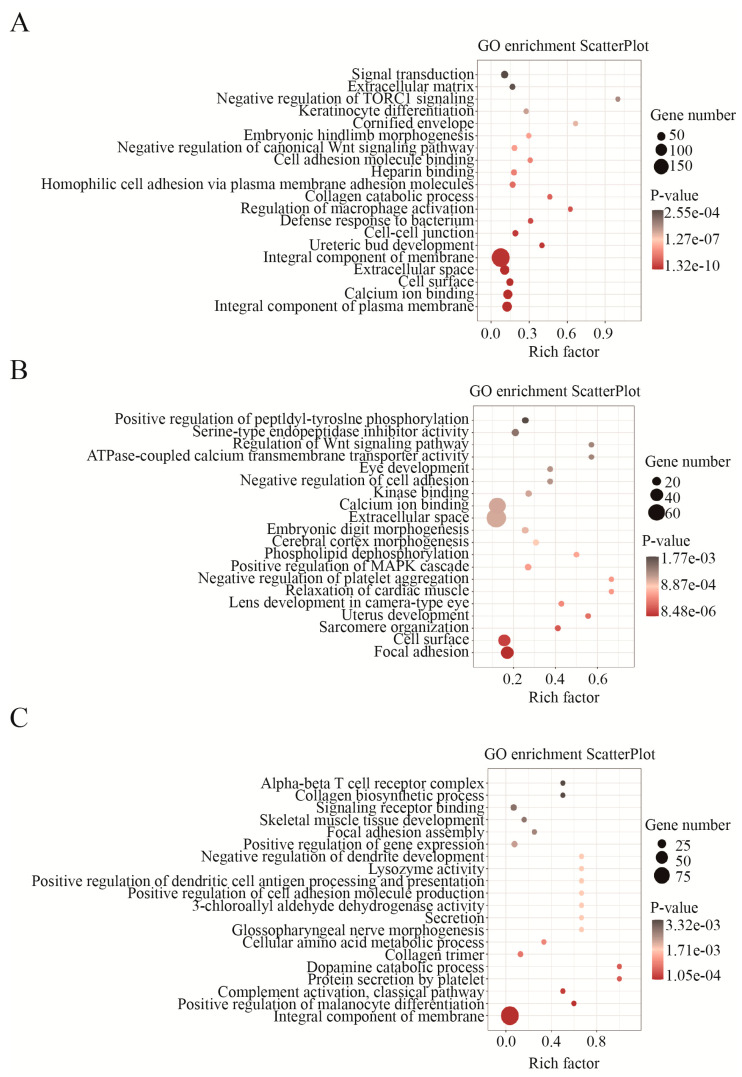
GO analysis of DEGs in HN10 and JDm10 induced the bursa of Fabricius. (**A**) Bubble charts showed GO enrichment (biological process) in the mutual DEGs of the bursa of Fabricius both induced by HN10 and JDm10. (**C**) Bubble charts showed GO enrichment (biological process) in the HN10-specific DEGs of the bursa of Fabricius. (**E**) Bubble charts showed GO enrichment (biological process) in the JDm10 specific DEGs of the bursa of Fabricius.

**Figure 4 viruses-15-00111-f004:**
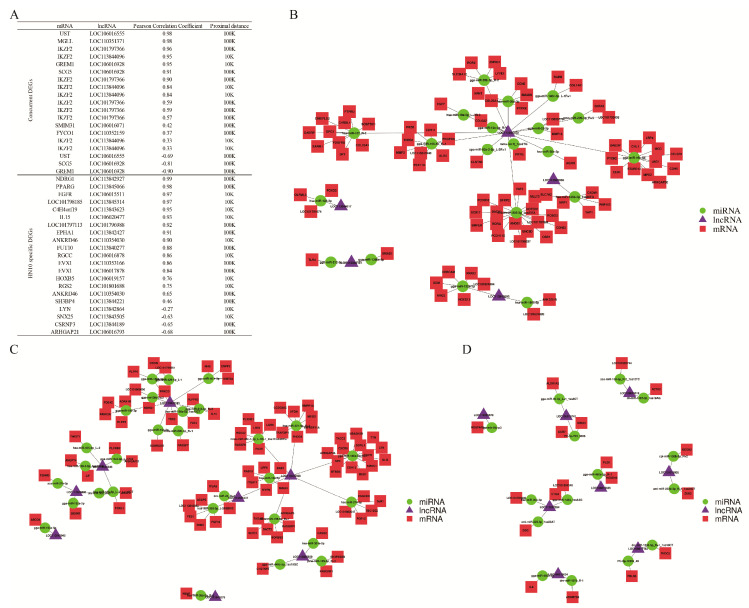
Network of differential expressed mRNA, lncRNA and miRNA in HN10 and JDm10 induced the bursa of Fabricius. (**A**) This table showed the interaction between differential expressed mRNA and lncRNA through the cis-regulatory module. (**B**–**D**) Network showed the interaction among differential expressed mRNA, lncRNA and miRNA special in HN10 (**B**) and JDm10 (**C**) as well as mutual in HN10 and JDm10 (**D**) through ceRNA modules.

**Figure 5 viruses-15-00111-f005:**
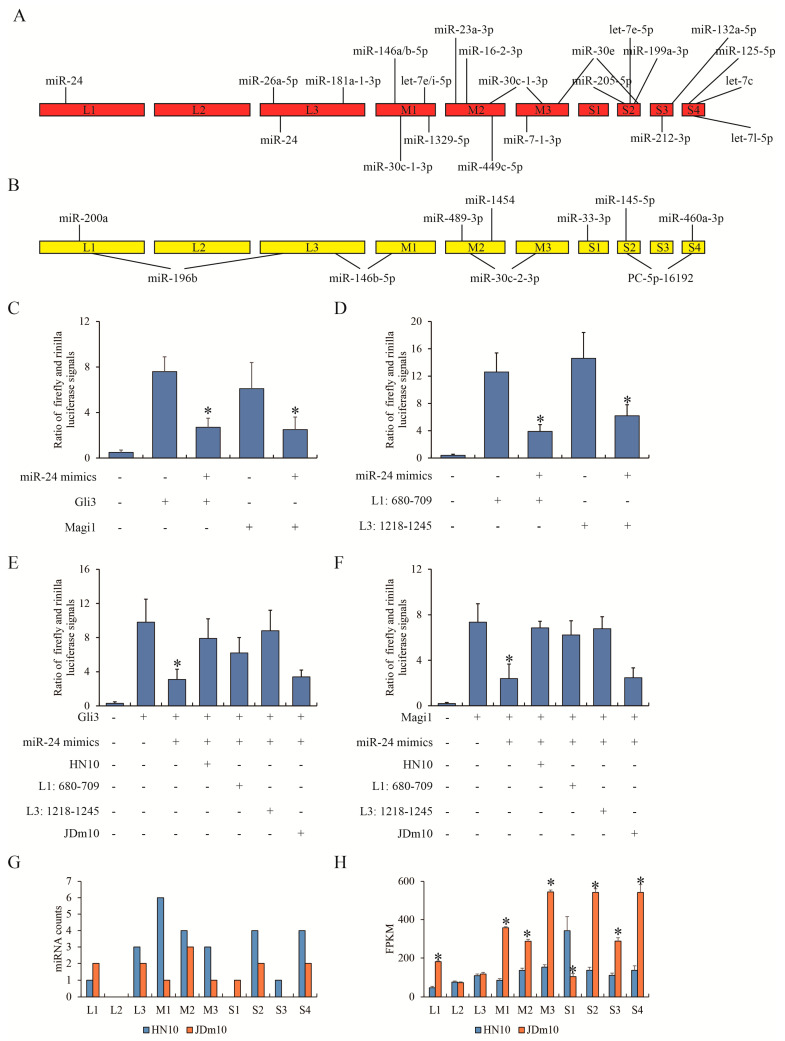
The interaction between miRNA and NDRV transcripts. (**A**) miRanda tool was used to predict the interaction between HN10 transcripts and HN10-induced differential expressed miRNAs compared to control. (**B**) miRanda tool was used to predict the interaction between JDm10 transcripts and JDm10-induced differential expressed miRNAs compared to control. (**C**) Dual luciferase reporter gene assay showed the effect of miR-24 on 3′ UTR of Gli3 and Magi1. miR-24 mimics and pGL3 firefly luciferase report vector cloned with 3′ UTR sequence of Gli3 or Magi1 were co-transfected into DEF cells. (**D**) Dual luciferase reporter gene assay showed the effect of miR-24 on the HN10 transcripts. miR-24 mimics and pGL3 firefly luciferase report vector cloned with L1 (680–709) and L3 (1218–1245) of HN10 transcripts were co-transfected into DEF cells. (**E**,**F**) Dual luciferase reporter gene assay showed the competitive role of HN10 transcripts in the transcriptional regulation of Gli3 (**E**) and Magi1 (**F**) by miR-24. “HN10” and “JDm10” were NDRV infection. “L1” and “L3” were synthetic nucleotides. (**G**) The counts of potential miRNA that can target to different fragments of HN10 and JDm10 transcripts. (**H**) The count of different fragments of HN10 and JDm10 transcripts in RNA-seq. Three individual replications of each experiment was performed. “*” indicated the statistical difference compared between previous group.

## Data Availability

The saw MS/MS data were deposited to Arrayexpress database assigned with the accession number E-MTAB-12115.

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
