# Peer review of "Distinct Whole Transcriptomic Profiles of the Bursa of Fabricius in Muscovy Ducklings Infected by Novel Duck Reovirus with Different Virulence"

_viruses, 2022, doi:10.3390/v15010111_

Round 1
Reviewer 1 Report
The authors conducted whole transcriptomic analysis to study the effects of two strains HN10 (virulent NDRV) and JDm10 (artificially attenuated NDRV) on the bursa of Fabricius of Muscovy ducklings. a large number of differential expressed genes (DEGs) of the bursa of Fabricius specially induced by HN10 and JDm10, and found that HN10 induced DEGs enriched in differentiation and development in multiple organs beyond JDm10. Before it can be accepted, several comments should be addressed.
1. More introduction of the two strains HN10 and JDm10 should be added to the Introduction section.
2. For Figure 1, there is great differences between HN10 and JDm10. However, no significant differences between JDm10 and NC was detected. Why?
3. The font size of Figure 3 is too small.
4. There is a mistake in Figure 4. The Figure 3 repeated in Figure 4.
5 The significance analysis in Figure 5G is missing.
Reviewer 2 Report
This is a very interesting manuscript. The competing endogenous RNAs across species can provide more ideas and clues for studying the infection by microorganism. Additionally, this research can also be served as a resource for further studying the molecular pathogenesis of Novel duck reovirus in Muscovy ducklings. I have no more questions.
Langauge needs to be polished by an English native speaker.
Author Response
This is a very interesting manuscript. The competing endogenous RNAs across species can provide more ideas and clues for studying the infection by microorganism. Additionally, this research can also be served as a resource for further studying the molecular pathogenesis of Novel duck reovirus in Muscovy ducklings. I have no more questions.
Langauge needs to be polished by an English native speaker.
A: We have asked the professional agency to polish the language.

Reviewer 3 Report
1. 2.3: two numbers, 108.2 and 106.4, are a formatting error in the paper ?
2.. 2.3: “The pathological changes of the bursa of Fabricius were observed ……..en for the consequent experiments” that describes the collection of samples, which should set in the 2.4 section.
3. The authors should introduce the background of miR-24 in Introduction, and discussion research significance of the change of miR-24 after infection by NDRV in duck.
